# Comparative Analysis of Bilateral Deficits in Elbow Flexion Strength: Functional vs. Analytical Assessment

Ignacio Pelayo-Tejo [1], Luis Chirosa-Ríos [1], Raquel Escobar-Molina [1], Amador García-Ramos [1,2,*], Indya del-Cuerpo [1], Ignacio Chirosa-Ríos [1] and Daniel Jerez-Mayorga [1,3]

1 Department of Physical Education and Sports, Faculty of Sport Sciences, University of Granada, 18011 Granada, Spain; nachopelayotejo@gmail.com (I.P.-T.); lchirosa@ugr.es (L.C.-R.); rescobar@ugr.es (R.E.-M.); delcuerpo@ugr.es (I.d.-C.); ichirosa@ugr.es (I.C.-R.); djerezmayorga@ugr.es (D.J.-M.)
2 Department of Sports Sciences and Physical Conditioning, Faculty of Education, Universidad Católica de la Santísima Concepción, 14138 Concepción, Chile
3 Exercise and Rehabilitation Sciences Institute, School of Physical Therapy, Faculty of Rehabilitation Sciences, Universidad Andres Bello, Santiago 7591538, Chile
* Correspondence: amagr@ugr.es

**Abstract:** Background: this study aimed to identify the influence of postural stability on upper-limb bilateral deficit (BLD), and to compare the assessment of strength generated during elbow flexion functionally vs. analytically in the dominant and nondominant arms. Methods: Twenty men participated in two sessions to evaluate the maximum isometric strength of elbow flexion. This evaluation was performed unilaterally with the dominant arm, unilaterally with the non-dominant arm, and bilaterally, both in the sitting position (SiP) and the standing position (StP). Results: The BLD when peak force was considered was lower for StP ($-6.44 \pm 5.58\%$) compared to SiP ($-10.73 \pm 6.17\%$) ($p = 0.007$). Regarding peak force, statistically significant differences were observed for comparisons between dominance ($p < 0.001$) and Position*Dominance ($p = 0.02$), but mean force differences were only observed for the dominance factor ($p < 0.001$). Greater mean and peak forces were always produced bilaterally compared to unilaterally ($p < 0.001$). Conclusions: a decrease in postural stability by performing elbow flexion exercises in a standing position accentuates BLD of peak force.

**Keywords:** isometric contraction; muscle strength dynamometer; bilateral index; neuromuscular

## 1. Introduction

Bilateral deficit (BLD) occurs when the sum of the strength produced by each limb individually is greater than the strength produced when both limbs apply force simultaneously [1,2]. This phenomenon has been observed in both the upper and lower limbs, regardless of whether the contraction is static or dynamic [3]. From a performance perspective, athletes should focus on training for optimal performance in the specific tasks of their sport, rather than aiming to equalize bilateral index values [4].

Since Henry and Smith described BLD in 1961 [5], many studies have attempted to elucidate the cause of BLD [6–8]. In 2016, Škarabot et al. [9] compiled possible mechanisms underlying the production of BLD. Specifically, they divided the classification into psychological, task-related, physiological, and neurophysiological factors. Despite the many mechanisms described, the most contrasted explanation for the origin of BLD is the neurophysiological factors; specifically, the inability to bilaterally activate fast-twitch muscle fibers at a maximal level [6–8]. BLD has been widely investigated in lower limbs [10–14]. In contrast, less scientific evidence exists for the upper limbs. The primary studies have focused on the finger [1,10–12,15], shoulder [16], and elbow [3,8,17–20] joints. According to Kotte et al. [21], the elbow joint is the most suitable joint for assessing upper limb strength. However, there is limited literature on BLD in this joint.

The presence of BLD has been confirmed in analytically assessed elbow flexion movements in healthy young adults [3,8,17,18]. Several studies agree on both the evaluation protocol of gesture and the fact that BLD is produced by a decrease in the electromyographic activity of the dominant limb [3,17,18]. These studies have shown that physiological and neurophysiological factors are responsible for BLD production in analytical gestures. However, other factors related to global and functional gestures, such as psychological and task-related factors, have been neglected.

With new technologies, it is possible to evaluate gestures both functionally and analytically. The assessment of functional movements can provide more practical conclusions regarding their application to daily life. This allows us to study other factors that trigger BLD in the upper limbs, such as the influence of postural stability or core action. Specifically, functional electromechanical dynamometry is a valid and reliable tool for performing functional assessments [22–26].

To the best of our knowledge, no study has functionally assessed elbow flexion movement. Therefore, new studies are needed to elucidate the effect of task-related factors, psychological factors, and other physiological factors on BLD. Therefore, the main objectives of this study were to (a) identify the influence of postural stability and core function modification on BLD and (b) compare the assessment of strength generated during elbow flexion functionally vs. analytically in the dominant (D) and nondominant (ND) upper-limbs.

## 2. Materials and Methods

A cross-sectional study design was used to compare the assessment of strength generated in elbow flexion both functionally and analytically, and to identify the influence of postural stability and core function modification on BLD. The maximum isometric strength of elbow flexion at a 90° angle was evaluated. This evaluation was performed unilaterally with the D arm, unilaterally with the ND arm, and bilaterally (BL) in the sitting position (SiP) and the standing position (StP).

Participants came to the laboratory twice. The first session was used for familiarization purposes and the second session was the main experimental session. A 48 h gap was left between sessions, allowing participants to recover after the effort made in the first session. None of the participants reported experiencing muscle soreness before the second test. Each session was scheduled at an identical time of the day and under similar environmental conditions (22 °C; 60% of humidity; 738 m above sea level).

### 2.1. Subjects

Twenty men volunteered to participate in the present study (age: 21.5 ± 2.9 years; height 178.5 ± 4.6 cm; body mass: 74.9 ± 8.1 kg; fat mass: 10.2 ± 4.6%; BMI: 23.5 ± 2.3 kg/m$^2$). The inclusion criteria for the study included physically active subjects who were right-handed and had not experienced any recent neuromuscular or musculoskeletal disorders.

The participants were informed verbally and in writing of the procedures before starting the study, and they provided written informed consent before participating in the study. The participants were instructed to come to the laboratory without exerting any physical exertion or ingesting alcohol, caffeine, or drugs for 24 h before the evaluation. The study protocol adhered to the tenets of the Declaration of Helsinki and was approved by the Institutional Review Board of the University of Granada (IRB approval: 928/CEIH/2019).

### 2.2. Procedures

2.2.1. Familiarization Session (Session 1)

In the first session, the participants visited the laboratory to receive instructions related to the evaluation process. This was followed by specific familiarization to teach participants how to perform maximal voluntary isometric contractions with the functional electromechanical dynamometer (FEMD). Each participant performed unilateral and bilateral isometric elbow flexion movements in both the SiP and StP. Each exercise was performed three times, with a rest between sets of 2 min.

During the session, a demographic and sports background survey was administered to all participants. In this survey, data, such as participants' age, sports they practiced, and time they spent practicing, were collected. Injuries and illnesses experienced before the study were also detailed.

In addition, anthropometric measurements were evaluated following the protocol designed by the International Society for the Advancement of Cineanthropometry [27]. Body mass was determined using a calibrated Tanita TBF-300 body composition analyzer goal setter calibrated scale (Tokyo, Japan) with an accuracy of $\pm 0.1$ kg. Height was measured using a Seca 123I stadiometer, $\pm 1$ mm accuracy. Arm perimeters and lengths were measured using a 0.5 cm-wide, 2 to 3 m-long, flexible but inextensible tape measure with a scale of 0.1 cm (Table 1).

**Table 1.** Upper limb lengths and circumferences.

|  | Forearm Length (cm) | Forearm Circumference (cm) | Arm Length (cm) | Relaxed Arm Circumference (cm) | Contracted Arm Circumference (cm) |
|---|---|---|---|---|---|
| Mean | 25.99 | 27.34 | 35.77 | 30.96 | 33.66 |
| Standard deviation | 1.02 | 1.74 | 1.64 | 3.27 | 3.20 |

2.2.2. Experimental Session (Session 2)

Forty-eight hours after the first session (i.e., familiarization), the participant returned to the laboratory for evaluation. The isometric strength was evaluated using a FEMD (Myoquality, Model M1, Granada, Spain) with a distance sensing accuracy of 3 mm and 100 g over the detected mass and a load range of 1–400 kg. The FEMD has a sampling frequency of 1 kHz and a velocity range from 0.05 ms$^{-1}$ to 2.80 ms$^{-1}$ [24].

The general warm-up began with global activation on a cycle ergometer at 60 rpm for 5 min, followed by joint mobility of the upper limbs. The specific warm-up consisted of performing 10 repetitions of bicep curl and 10 repetitions of wrist curl with 2 kg dumbbells for both arms. The order in which the exercises were performed, the condition (SiP or StP), and the arm evaluated (unilateral D, unilateral ND, or BL) were randomized. Following the recommendations of Baldwin et al. [28], a duration of 8 s of maximal isometric contraction was set for each trial, with a 3 min rest before performing the next trial. This procedure was repeated for each of the six evaluated positions. The 90° measurement was ensured using a manual goniometer.

All three assessments performed in the seated condition were performed on a Scott bench. The seat height was adjusted for each participant before beginning the test. For the BL evaluation, the arms were resting on the Scott bench with a separation between them equal to the width of the shoulders. The ventral portion of the thorax and posterior portion of the arm were placed in contact with the edge of the Scott bench. The grip was performed in a supine position with the shoulders in a neutral position and the elbows flexed at 90° (Figure 1).

For the unilateral seated assessment, the initial position was the same as that adopted for the BL assessment, with the only difference being that the non-performing arm was placed in a stretched (i.e., full extension) and relaxed way. The elbow joint of the performing arm was placed at 90° of flexion, and both shoulders were kept in a neutral position (Figure 2).

BL assessment was performed in the standing position at 0° of plantar flexion. The feet were placed parallel to each other with a separation between them equal to the width of the shoulders. The knees were semi-flexed. The grip was performed in a supine position while the shoulders were in a neutral position and the elbows were flexed at 90°. Participants were instructed to keep their heads straight with their eyes fixed forward throughout the isometric performance (Figure 3).

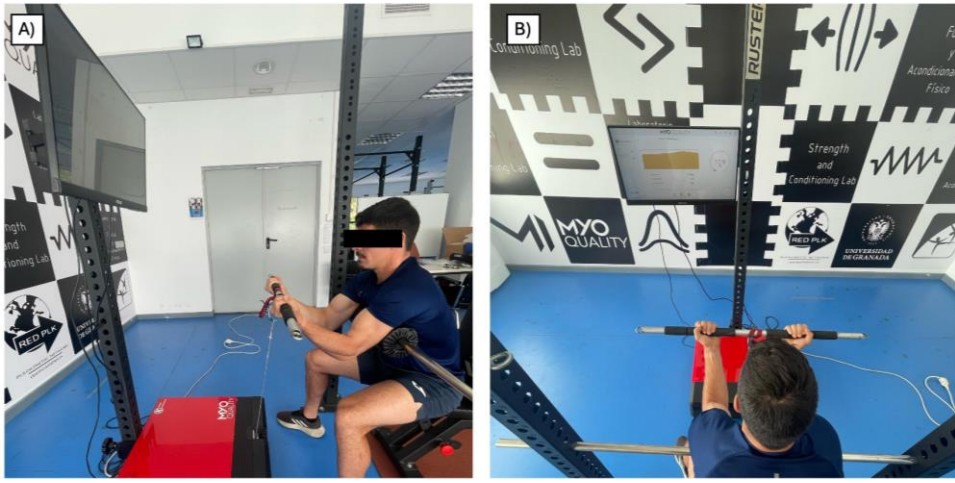

**Figure 1.** In the sitting position, bilateral elbow flexion is performed at a 90° angle: (**A**) side view, (**B**) back view.

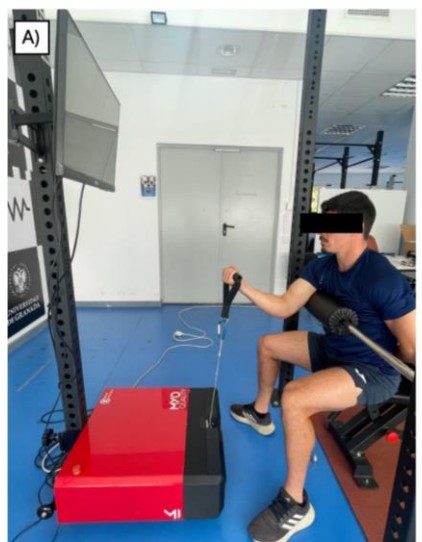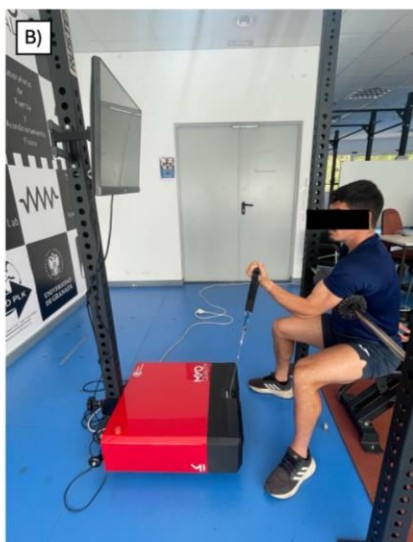

**Figure 2.** In the sitting position, perform unilateral elbow flexion at a 90° angle: (**A**) right arm, (**B**) left arm.

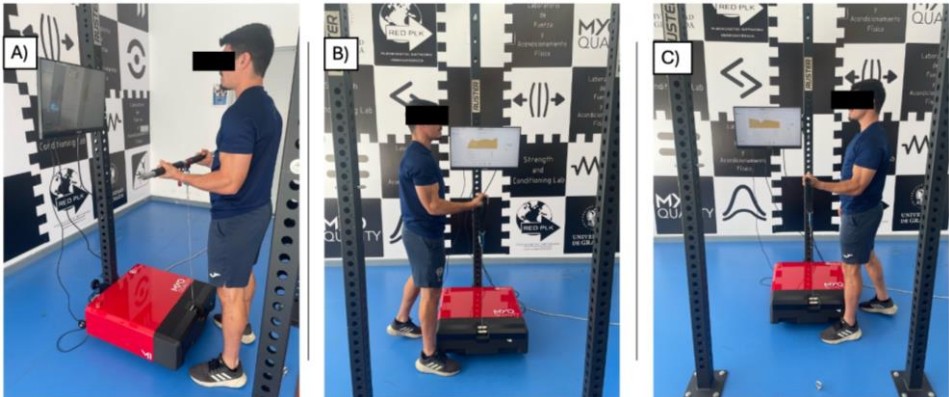

**Figure 3.** Standing upright, performing: (**A**) bilateral elbow flexion with both arms bent at a 90° angle, (**B**) unilateral right elbow flexion with the right arm bent at a 90° angle, and (**C**) unilateral left elbow flexion with the left arm bent at a 90° angle.

The unilateral assessment was performed in the standing position at 0° of plantar flexion with the opposite leg in front of the executing arm. The knees were semi-flexed. The

grip was performed in a supine position while the shoulders were in a neutral position and the performing elbow was flexed at 90°. The non-performing arm remained stretched (i.e., full extension) and relaxed. As in the BL condition, the participants were instructed to keep their head straight with their eyes fixed forward throughout the isometric performance.

### 2.2.3. Bilateral Deficit Calculation

The bilateral index (BI), which is the ratio of the percentage of BLD, was calculated from the left and right unilateral and BL strength tests. The BI was determined using the following equation [29]:

$$BI(\%) = 100 \times \frac{B}{(\text{Unilateral D} + \text{Unilateral ND})} - 100$$

A BI equal to zero indicates no differences between BL and unilateral contractions. A negative BI value indicates that the performance during BL contractions is lower than the combined performance during unilateral contractions, i.e., BLD. On the other hand, a positive BI value indicates that the performance during BL contractions is superior to that during unilateral contractions. Therefore, the opposite phenomenon, called BL facilitation, occurs.

### 2.3. Statistical Analyses

Descriptive data are presented as mean and standard deviation (SD). Normal distribution of the data (Shapiro–Wilk test) and homogeneity of variances (Levene test) were confirmed ($p > 0.05$). The *t*-test for related samples was performed to compare the BLD values produced with SiP vs. those produced with StP. For the main analysis, repeated-measures analysis of variance (ANOVA) was conducted with Bonferroni's post hoc analysis. The Greenhouse–Geisser correction was used when the Mauchly sphericity test was violated. Omega squared ($\omega^2$) was calculated for the ANOVA, where the values of effect sizes of 0.01, 0.06, and >0.14 were considered small, medium, and large, respectively [30]. Statistical significance was set at the $p < 0.05$ level. The JASP statistics package (version 0.15) was used for statistical analyses.

### 3. Results

No statistically significant differences were found between the BLD generated in SiP vs. StP for the mean forces ($p = 0.155$). On the other hand, differences were observed in the BLD of the peak force ($p = 0.007$). In addition, a smaller BLD was observed in the movements of StP than SiP (Figure 4).

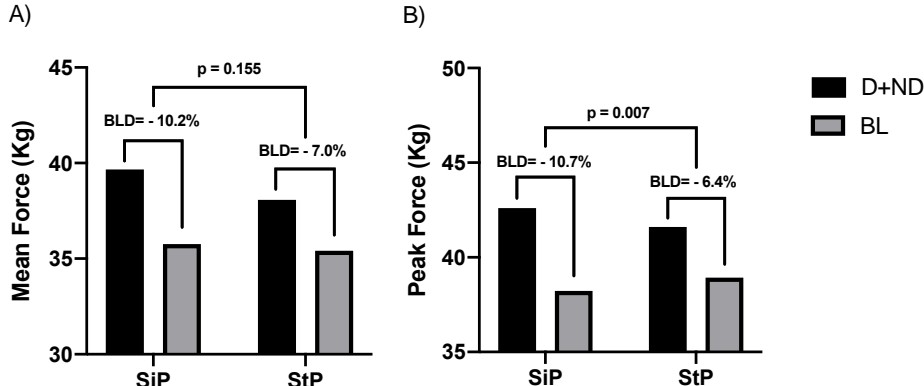

**Figure 4.** Mean force (**A**) and peak force (**B**) (kg) obtained during the performance of the isometric elbow flexion movement, both bilaterally and unilaterally (dominant + nondominant) for sitting position (SiP) and the standing position (StP).

On the other hand, statistically significant differences with a "large" effect size were observed for comparisons between dominance ($p < 0.001$; $\omega^2 = 0.673$) and mean force. Post hoc analysis using Bonferroni correction revealed that when comparing unilateral D movement to unilateral ND movement, there were no statistically significant differences (mean difference = 0.210; $p = 1.00$). In contrast, this post hoc analysis revealed that the comparisons between dominance differed when comparing D + ND vs. BL (mean difference = 3.28; $p < 0.001$) in favor of D + ND. Regarding peak force, statistically significant differences with a "large" effect size were observed for comparisons between dominance ($p < 0.001$; $\omega^2 = 0.638$), and statistically significant differences with a "small" effect size were observed for comparisons between Position*Dominance ($p = 0.02$ $\omega^2 = 0.002$). Post hoc analysis of dominance revealed that when comparing D with ND, there were no statistically significant differences (mean difference = 0.277; $p = 1.00$).

In contrast, this post hoc analysis revealed that comparisons between dominance differed when comparing D + ND vs. BL (mean difference = 3.53; $p < 0.001$). Post hoc analysis of Position*Dominance comparisons shows that there is no statistically significant difference between SiP.D vs. StP.D (mean difference = 0.870; $p = 1.00$); SiP.D vs. SiP.ND (mean difference = 0.642; $p = 1.00$); SiP.D Vs StP.ND (mean difference = 0.782; $p = 1.00$); StP.D Vs SiP.ND (mean difference = $-0.228$; $p = 1.00$); StP.D vs. StP.ND (mean difference = $-0.088$; $p = 1.00$); SiP.ND vs. StP.ND (mean difference = 0.140; $p = 1.00$); SiP.D + ND vs. StP.D + ND (mean difference = 1.010; $p = 1.00$); StP.D + ND vs. SiP.BL (mean difference = 3.377; $p = 0.064$); StP.D + ND vs. StP.BL (mean difference = 2.672; $p = 0.093$) and SiP.BL vs. StP.BL (mean difference = $-0.705$; $p = 1.00$). On the other hand, statistically significant differences were found when comparing SiP.D + ND vs. SiP.BL (mean difference = 4.387; $p < 0.001$) and SiP.D + ND vs. StP.BL (mean difference = 3.683; $p = 0.026$).

For both mean and peak force, the post hoc analyses comparing unilateral movements (i.e., D or ND) vs. D + ND or BL was significantly associated with the latter ($p < 0.001$) (Table 2).

**Table 2.** Comparison of the strength generated by comparing different variables for isometric elbow flexion.

| | Variable | D | ND | D + ND | B | Repeated Measures ANOVA |
|---|---|---|---|---|---|---|
| Mean force (kg) | SiP | 20.08 (4.15) | 19.59 (4.59) | 39.67 (8.61) | 35.77 (8.81) | Position $F = 2.030$; $p = 0.171$; $\omega^2 = 0.003$ |
| | StP | 19.01 (4.10) | 19.08 (3.58) | 38.08 (7.46) | 35.42 (7.74) | Dominance $F = 342.890$; $p < 0.001$; $\omega^2 = 0.673$ Position*Dominance $F = 1.397$; $p = 0.260$; $\omega^2 = 0.001$ |
| Peak force (kg) | SiP | 21.63 (4.39) | 20.99 (4.76) | 42.61 (9.02) | 38.23 (9.42) | Position $F = 0.263$; $p = 0.614$; $\omega^2 = 0.000$ |
| | StP | 20.76 (4.15) | 20.85 (4.00) | 41.60 (7.97) | 38.93 (7.86) | Dominance $F = 361.910$; $p < 0.001$; $\omega^2 = 0.638$ Position*Dominance $F = 4.170$; $p = 0.02$; $\omega^2 = 0.002$ |

Abbreviations: D = dominant, ND = nondominant, D + ND = dominant + nondominant, B = bilateral, SiP = sitting position, StP = standing position. Data are presented as mean (standard deviation).

## 4. Discussion

The main objectives of this study were to (a) identify the influence of postural stability and core function modification on BLD, and (b) compare the assessment of strength generated during elbow flexion functionally vs. analytically and D vs. ND. Based on these findings, the main results of this study were that (a) greater BLD occurs in movements performed while seated compared with gestures performed while standing. Furthermore, it has been shown that (b) the change from standing to seated does not modify the mean and peak force production during isometric elbow flexion. In contrast, dominance did affect

mean and peak force, and position together with dominance (i.e., Position*Dominance) only affected peak force.

The results obtained in this study (mean force BLD = −10.2% and peak force BLD = −10.8%) agree with those published by Scott P. McLean et al. (2006) (BLD = −10.2%). In this study, as in the present study, isometric elbow flexion at 100% maximal strength was executed on a Scott bench with a 90° angulation. This BLD is caused by decreased motor unit recruitment during BL movement, inhibition of neural circuitry [3], and decreased activation of the D limb [15,17,18].

Regarding the results obtained in BLD produced in StP, no previous studies employ a methodology similar to that used in this study. In general, BLD produced in different movements and muscle groups (i.e., lower limbs vs. upper limbs/squat vs. handgrip) are usually compared. There is a theory that postural or joint stability may influence BLD expression [11,31]. Specifically, this theory indicates that greater BLD occurs in multi-joint movements than in mono-joint gestures. This is largely due to postural stability and core activity.

When comparing the BLD produced in the present study for the same gesture but in different positions (i.e., seated vs. standing), we found that, contrary to what was described above, the movement that involves larger body structures (standing) and requires greater postural stability and core activation [32] has a lower BLD than the movement performed in a more analytical manner (seated). Therefore, caution should be exercised when interpreting force assessments and conclusions, as it is very difficult to accurately compare methodologies, measurement instruments, force application angles, rest duration, and variables that may influence the manifestation of such forces [1,21].

When comparing force production during movement, changes in position from SiP to StP did not modify mean force production or peak isometric elbow flexion. In contrast, dominance affects mean and peak forces, whereas Position*Dominance only affects peak forces. To the best of our knowledge, no previous study has investigated force production during isometric elbow flexion in different positions (i.e., SiP vs. StP). In contrast, some studies have evaluated the influence of position on other movements [32–34]. Saeterbakken et al. (2013) indicated in their research that the 1 RM strength and lifting time were the same in SiP compared with StP when performing a barbell shoulder press [34].

In terms of force production between D + ND vs. BL, we found differences in peak force when comparing the SiP.D + ND gesture to the BL gesture, regardless of whether it was in SiP or StP. In SiP movement, force production is higher because greater stability is achieved in execution facilitated by a better support base and a larger number of contact points [34]. In addition, force production in D + ND is higher than that in BL due to the action of psychological, task-related, physiological, and neurophysiological factors [6]. On the other hand, when we compared the sum of D + ND in StP with the BL movements, we did not find any statistically significant differences. The main reason for this result is that having a smaller base of support than that in SiP makes the force production somewhat lower than that in StP [34] and equalizes the results to those obtained in the BL movement.

According to the results obtained after conducting this research, if we compare a gesture performed unilaterally (D or ND) with a gesture performed bilaterally (D + ND or BL), regardless of the position (SiP or SiP), we observe that force production is always higher in gestures performed bilaterally than in gestures performed unilaterally. These results cannot be justified by any specific article because, to our knowledge, none of them make such a comparison. Despite this, we can, however, compare tables of results and figures from different articles, which seem to be in line with our results [6]. This finding is closely related to the conclusions obtained by Janzen et al. (2006) that BL strength training generates greater strength gains than unilateral strength training [31].

Finally, regarding dominance, it was observed that among the unilateral movements when comparing D with ND ($p = 0.210$), the results obtained agree with those previously published by Kotte et al. (2018) [21]. In that review, the authors concluded that there was no difference between elbow flexion with the D arm and elbow flexion with the ND arm ($p = 0.312$).

The present study was limited by the non-use of electromyographic control in both the lower limbs and the core area to control possible modifications to postural and core stability when modifying the type of movement. Although it was already justified by previous studies, the use of this tool could have provided more robust conclusions.

Additionally, we note that the present results are limited to only healthy resistance-trained men; therefore, these results cannot necessarily be generalized to other populations. Future research should evaluate both BLD and the influence of position on the manifestation of strength in individuals in different population groups (sedentary, with some type of pathology, advanced age, etc.). Furthermore, the neuromuscular patterns of exercises with higher and lower stability requirements and intensities should be investigated. Finally, it would be interesting to investigate how core strength improvement affects BLD.

## 5. Conclusions

A decrease in postural stability and an increase in core function (i.e., StP) only affected the BLD of peak force, not that of mean force, during isometric elbow flexion. On the other hand, no difference was observed when comparing unilateral D vs. ND movement, with no influence of position or dominance observed when comparing movement D with ND. A greater peak force was also found in SiP D + ND than in BL in both SiP and StP. Finally, unilateral movements (i.e., D or ND) produced lower force values than BL movements (i.e., D + ND or BL).

Bilateral strength training reduces BLD, whereas unilateral strength training increases BLD [2]. In predominantly bilateral sports, lower BLD or greater bilateral facilitation is associated with enhanced performance. Conversely, in sports where changes in direction or unilateral movements are predominant, greater BLD is associated with greater performance [13]. Therefore, it is up to each coach to decide what is more interesting to improve the performance of their athletes.

The findings of this study have significant implications for both sports performance and rehabilitation contexts. Understanding how postural stability and core function influence BLD in elbow flexion can inform the design of more effective training and rehabilitation programs. For athletes, training protocols can be tailored to either reduce or exploit BLD depending on the specific demands of their sport, such as optimizing bilateral strength for symmetrical sports or enhancing unilateral strength for sports requiring asymmetrical movements. In rehabilitation, practitioners can use these insights to develop targeted interventions that improve functional strength and stability in patients recovering from upper limb injuries. Additionally, the comparison between functional and analytical assessments provides valuable information for selecting the most appropriate evaluation methods in clinical and sports settings, ensuring that strength assessments are both relevant and transferable to real-world tasks.

The knowledge obtained from this research allows us to determine BLD in the upper extremities through functional movement. In addition, it makes it possible to determine the influence of position on BLD, thus discriminating which method presents a lower BLD.

**Author Contributions:** Conceptualization, I.P.-T., I.C.-R., L.C.-R. and D.J.-M.; methodology, I.P.-T., I.C.-R., L.C.-R. and D.J.-M.; software, I.P.-T., I.C.-R., L.C.-R. and D.J.-M.; validation, I.P.-T., I.C.-R., L.C.-R. and D.J.-M.; formal analysis, I.P.-T., I.C.-R., L.C.-R. and D.J.-M.; investigation, I.P.-T., R.E.-M., A.G.-R. and I.d.-C.; resources, I.P.-T., R.E.-M., A.G.-R. and I.d.-C.; data curation, I.P.-T., I.C.-R., L.C.-R. and D.J.-M.; writing—original draft preparation, I.P.-T.; writing—review and editing, I.P.-T., R.E.-M., A.G.-R., I.d.-C., I.C.-R., L.C.-R. and D.J.-M.; visualization, I.P.-T., I.C.-R., L.C.-R. and D.J.-M.; supervision, I.C.-R., L.C.-R. and D.J.-M.; project administration, I.P.-T., I.C.-R., L.C.-R. and D.J.-M.; funding acquisition, I.P.-T., I.C.-R., L.C.-R. and D.J.-M. All authors have read and agreed to the published version of the manuscript.

**Funding:** This work was supported by the FEDER/Junta de Andalucía/Project (B-CTS-184-UGR20) and Consejería de Universidad, Investigación e Innovación (A.SEJ.227.UGR23).

**Institutional Review Board Statement:** The study protocol adhered to the tenets of the Declaration of Helsinki and was approved by the University of Granada Institutional Review Board (IRB approval: 928/CEIH/2019).

**Informed Consent Statement:** Informed consent was obtained from all subjects involved in the study.

**Data Availability Statement:** The raw data supporting the conclusions of this article will be made available by the corresponding author on request.

**Acknowledgments:** This paper will be part of Ignacio Pelayo-Tejo Doctoral Thesis performed in the Biomedicine Program of the University of Granada.

**Conflicts of Interest:** The authors declare no conflicts of interest.

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
