# Peer review of "Comparative Analysis of Bilateral Deficits in Elbow Flexion Strength: Functional vs. Analytical Assessment"

_applsci, doi:10.3390/app14177808_

Round 1

Reviewer 1 Report

Comments and Suggestions for Authors

Here are my comments on your manuscript:

1. why the shift in focus to psychological and task-related factors is necessary?

2. add an explanation of how postural stability and core function might impact BLD (add some literature citations)

3. add an indepth description on why functional assessment of elbow flexion is important and how it will contribute to the broader understanding of BLD

4. What are the limitations of using functional electromechanical dynamometry?

5. how were postural stability and core functions manipulated and/ or assessed during the study?

6. what is the justification on using the Scott bench

7. describe the randomization process to ensure transparency with reproducibility

8. a description or reference for each piece of equipment used is required

9. how did you account for / minimize the potential impact of altitude (738m above sea level) on participants performance?

Reviewer 2 Report

Comments and Suggestions for Authors

The paper is well written and depicts important aspects of postural stability and biomechanics related to unilateral/bilateral elbow motions.

But the paper could be improved regarding the following aspects:

Results and statistical analysis

Here are some suggestions for improving the statistical analysis:

While the study uses repeated-measures ANOVA, it could benefit from multivariate approaches that account for the potential interaction effects between multiple dependent variables (e.g., peak force, mean force) and the independent variables (e.g., position, dominance). Multivariate analysis of variance (MANOVA) could provide a more comprehensive understanding of how these factors interact.

The reporting of effect sizes could be expanded to include partial eta-squared or Cohen’s d for more direct comparisons with other studies.

Control for Potential Confounding Variables-The study could include covariates in the analysis to control for potential confounding factors, such as participants' baseline strength, training history, or fatigue levels. Including these as covariates in an ANCOVA (Analysis of Covariance) could help isolate the effects of the primary independent variables.

Clarity in Reporting Statistical Results: The statistical results could be presented more clearly, with a more detailed breakdown of how interaction effects (e.g., Position * Dominance) were interpreted. Including confidence intervals for means and effect sizes would also provide a more nuanced understanding of the variability and reliability of the results.

Discussion section 

The authors should emphasize the importance of their research findings that can be further investigated and applied in practice (sports, physiotherapy, etc.).
